# Effectiveness of Elastic Therapeutic Tape in Reducing Edema, Pain and Trismus following Surgery for Facial Fractures: A Systematic Review and Meta-Analysis

**DOI:** 10.3390/jcm13040997

**Published:** 2024-02-09

**Authors:** Rebeca Valeska Soares Pereira, Sandra Lúcia Dantas de Moraes, João Luiz Gomes Carneiro Monteiro, Ana Cláudia Amorim Gomes, Eduardo Piza Pellizzer, Belmiro Cavalcanti do Egito Vasconcelos

**Affiliations:** 1Oral and Maxillofacial Surgery, Department of Oral and Maxillofacial Surgery and Traumatology, University of Pernambuco, Recife 50100-130, Brazil; rebeca.valeska@upe.br (R.V.S.P.); anacagomes@upe.br (A.C.A.G.); 2Faculty of Dentistry, Department of Oral Rehabilitation, University of Pernambuco, Recife 50100-130, Brazil; sandra.moraes@upe.br; 3Massachusetts General Hospital and Harvard Medical School, Boston, MA 02155, USA; jlmonteiro@mgh.harvard.edu; 4Araçatuba Dental School, Department of Dental Materials and Prosthodontics, São Paulo State University (UNESP), Araçatuba 16015-050, Brazil; ed.pl@uol.com.br

**Keywords:** Kinesio taping, facial trauma, face surgery, systematic review

## Abstract

Facial fractures cause postoperative morbidity, including edema, pain, and trismus. Elastic therapeutic tapes are used for optimizing recovery. **Background:** The aim of the present systematic review and meta-analysis was to evaluate the effectiveness of elastic tape Kinesio taping (KT) in reducing postoperative morbidity in facial fractures surgeries. **Methods:** A systematic review was conducted in accordance with the PRISMA guidelines. Searches were conducted in the Cochrane, Medline, Scopus, Embase and Web of Science databases using a pre-established search strategy. **Results:** A total of 811 studies were retrieved after the duplicates were removed, and only randomized clinical trials were included. Eight trials, involving 319 participants, were deemed eligible. One study solely investigated the effect on edema, while the others analyzed at least two of the variables of interest. Results from two RCTs, where qualitative analysis was applicable, suggest a potential reduction in edema in the KT group compared to the control group on the second (RR −0.55, 95% CI −0.89 to −0.22; *p* = 0.01; I^2^ = 0%) and third postoperative days (RR −0.71, 95% CI −1.01 to −0.40; *p* < 0.00001; I^2^ = 0%). **Conclusions:** KT is effective in controlling postoperative edema following surgery for facial fractures. However, the effects on pain and trismus should be explored further in studies with standardized methods.

## 1. Introduction

Edema (swelling), pain and trismus (limited mouth) are common signs and symptoms in the postoperative period of victims of facial trauma [1]. Pain is typically brief, with maximum intensity in the immediate postoperative period, whereas facial edema and trismus are most evident between 48 and 72 h after surgery, diminishing gradually over the course of days/weeks [1,2].

These signs and symptoms affect the quality of life and delay patient recovery [3,4]. The literature reports several methods for reducing edema in the postoperative period of maxillofacial surgery, such as steroids in the pre-, intra- and postoperative periods [5,6,7], proteolytic enzymes [8], laser therapy [9], cryotherapy or manual lymphatic drainage [10,11].

Compression bandages are considered a low-cost method [12]. Kinesio taping was developed in the 1970s by Japanese therapist and professor Kenzo Kase. The material used for this technique is made of cotton and has elasticity of up to 140%. The elastic effect operates on the elevation of the skin surface, which increases the space between the dermis and fascia, reducing lymphatic retention and the discomfort that emerges due to the pressure resulting from the occurrence of lymphedema [13,14].

The literature offers a number of randomized clinical trials involving Kinesio taping to diminish morbidity in the maxillofacial region in the immediate postoperative phase of orthognathic surgery [15], surgically assisted rapid maxillary expansion [16], third-molar surgery [17,18,19,20,21,22], mandibular fractures and [23] fractures of the orbital–zygomatic complex [24]. However, no systematic reviews have been conducted on the use of this method following surgery for facial trauma.

Therefore, the aim of the present study was to perform a systematic review of the literature to determine the effectiveness of KT in reducing postoperative signs and symptoms following surgery for facial fractures that compromise patient wellbeing and increase the degree of morbidity.

## 2. Materials and Methods

### 2.1. Study Design

The protocol for this study was previously registered in the International Prospective Register of Systematic Reviews (CRD42023442659). The present literature review was conducted following the revised Preferred Reporting Items for Systematic Reviews and Meta-Analyses (PRISMA statement) [25].

### 2.2. Inclusion Criteria

The PICO structure was employed to define the inclusion criteria. All eligible randomized clinical trials that assessed male and female patients aged 18 years or older having been submitted to surgical procedures for the reduction and fixation of facial fractures (Population—P), treated in the postoperative period with Kinesio taping (Intervention—I) and compared to a group in which no elastic device was employed (Comparation—C), with the assessment of edema, pain and trismus (Outcome—O) were considered for inclusion. 

### 2.3. Exclusion Criteria

The exclusion criteria were case reports, articles not published in journals recognized in the databases consulted, narrative and systematic reviews, letters to the editor, animal studies and in vitro studies. No restrictions were imposed with regards to language or year of publication.

### 2.4. Search Strategy and Article Selection

Terms related to the topic of interest were combined using Boolean operators in searches of the Cochrane Central Register of Controlled Trials, Medline (Pubmed), Scopus, Embase and Web of Science databases in a two-month period (May to June 2023). Searches were conducted using MeSH terms as an appropriate search strategy. A hand search was also conducted of periodicals in the field for articles not yet indexed in the databases. 

The records retrieved during the searches were imported to the Rayyan application for the initial screening of titles and abstracts and removal of duplicates. Titles and abstracts were analyzed by two reviewers (RVSP, JLGCM) working independently for the preselection of articles for full-text analysis based on the eligibility criteria. Articles that met the eligibility criteria were included in the review.

### 2.5. Summary Measures

The Review Manager software (RevMan version 5.4.1; The Nordic Cochrane Centre, The Cochrane Collaboration, Copenhagen, Denmark, 2014) was used for meta-analysis and to create forest plots of the edema variable involving studies that described adequate mean and standard deviation values for the quantitative analysis [26,27]. The Mantel-Haenszel method was employed. Risk difference (RD) with a 95% confidence interval (CI) was used to determine the change facial measurements between the preoperative period (T-1) and the second and third days of the postoperative period. RD was considered significant when the *p*-value was <0.05. 

If statistically significant heterogeneity had been found (*p* < 0.10), a random-effects model would have been used to determine the effects of treatment. As this was not the case, however, a fixed-effects model was used. Heterogeneity was determined using the Q (X2) method, with a calculation of the I^2^ value, which was used to analyze heterogeneity; an I^2^ value higher than 75% (0–100%) was considered indicative of relevant heterogeneity [27]. Two of the articles included in the review were eligible for the quantitative analysis—both for the assessment of edema on the second and third days postoperatively compared to preoperative facial measurements. The other articles were not included in the quantitative analysis due to the incompatible presentation of the data (only means) or the absence of necessary data.

### 2.6. Data Extraction

Two reviewers (RVSP, JLGCM) independently performed the extraction of relevant data from the articles. The main variable considered was edema and results of the pre-established linear measurements performed in each study were demonstrated by the sum of the means (in mm). Pain (analyzed using the visual analogue scale from zero to 10) and trismus (distance between maxillary and mandibular central incisors) were the secondary outcomes. The following data were also extracted from the articles: authors, year of publication, country in which the study was developed, inclusion criteria, postoperative follow-up time, age and sex of the participants, sample size per group, KT type and technique, medications used and outcomes. 

### 2.7. Appraisal of Risk of Bias

Risk of bias of the randomized clinical trials included in the present review was appraised using the Revised Cochrane risk-of-bias tool for randomized trials (RoB 2.0) [28], which is currently recommended by the Cochrane Collaboration. Two independent reviewers (RVSP, JLGCM) performed the appraisal considering each category and outcome. Cases of a divergence of opinion between the reviewers were resolved by consulting a third reviewer (BCEV) with ample experience in this type of analysis.

## 3. Results

### 3.1. Article Selection

The article selection process is displayed in the flowchart (Figure 1—Flowchart of article selection process). The searches of the databases resulted in the retrieval of 960 records: 377 in Medline/PubMed, 270 in Web of Science, 194 in Embase, 89 in the Cochrane Library and 30 in Scopus. One hundred forty-nine articles appeared in more than one database, and the duplicates were removed. After the analysis of the titles and abstracts, 25 articles were submitted to full-text analysis, 17 of which were excluded for being cases of maxillofacial surgery not related to facial trauma (13 cases of third molar surgery and 4 cases of orthognathic surgery). Thus, eight articles were included in the present systematic review [23,24,26,27,29,30,31,32].

### 3.2. Characteristics of Studies Included

All studies included in the present systematic review were randomized clinical trials. Table 1 displays the general characteristics of the studies, such as authors, year of publication, country in which the study was conducted, region of the fracture, follow-up time as well as the age, sex and number of participants. A total of 319 patients were included: 161 in the intervention groups and 158 in the control groups. Men predominated in the overall sample. The age range was 18 to 75 years; age was not specified in one of the studies [31]. In terms of the region of the facial fracture, three studies only considered fractures of the mandible [23,27,30], three only considered fractures of the zygomatic–orbital complex [26,29,31] and two addressed both types of fractures [24,32].

Only four studies used the same brand of kinesiological tape [23,24,29,31]. Variability was also found with regards to the Kinesio taping technique. Six different techniques were described (Table 2), with variations in terms of the shape and distribution of the elastic strips. All studies maintained Kinesio therapy for five days in the postoperative period.

Most of the articles included in the present systematic review failed to describe the medications used in the preoperative period. Ristow, Pautke et al. (2013) and Ristow et al. (2014) administered a single dose of 2000 mg/1000 mg ampicillin/sulbactam, whereas Krishnamurthy et al. (2021) [26] used 1000 mg hydrocortisone half an hour prior to anesthetic induction. Several medications were used in the postoperative period, such as 1000 mg [23,24,30] and 750 mg [27] paracetamol, 600 mg ibuprofen [23,24], 2000 mg/1000 mg ampicillin/sulbactam [23], 50 mg diclofenac [31,32], 1 g ceftriaxone [30] and 500 mg amoxicillin [27].

The studies were assessed with regards to the feasibility of data synthesis, but meta-analysis was not possible for two of the three outcomes due to the heterogeneity of the data. The sources of heterogeneity were research groups that likely used the same population in different studies [23,24,31], different Kinesio taping methods, different follow-up period and different ways of analyzing the results.

### 3.3. Edema

Edema was measured in all studies using linear measurements between predefined anatomic reference points (either five [23,24,26,29,31,32] or three [27,30] reference lines). No preoperative measurements were made in the studies conducted by Deleme, Aljubory (2021) and Tyndorf et al. (2016). The measurement interval in the postoperative period was variable, but follow-up on the first, second, third and seventh days was performed in most studies [23,24,29,31]. Differences were found in the reduction in edema in the intervention and control groups, demonstrating a possible effect of Kinesio taping (Table 3). As heterogeneity was nonsignificant between two studies [26,27] with regards to the edema variable (changes in the first 72 h of the postoperative period taking the baseline value measured in the preoperative period as reference), meta-analysis was performed using a random-effects model (Figure 2 and Figure 3).

### 3.4. Trismus

Two studies did not assess trismus in patients submitted to facial surgery [30,32]. Maximum mouth opening was determined by the maximum distance between the maxillary and mandibular incisors using calipers and expressed in centimeters. Greater mouth opening capacity was generally found in the group with Kinesio taping compared to the control group, beginning on the second day of the postoperative period. Krishnamurthy et al. (2021) [26] described equal maximum mouth opening measurements in both groups.

### 3.5. Pain

Tyndorf et al. (2016) [32] did not assess pain. In the other studies, pain was measured using the visual analogue scale, on which 1 corresponds to minimal pain reported and 10 corresponds to maximum pain. Analysis of the data revealed that peak pain was reported in the first 24 h after the surgical procedure in most cases, and the patients in both groups scored the pain as mild to moderate (VAS score < 5). Although lower pain scores were found in the groups submitted to Kinesio taping compared to the control groups, the difference was not statistically significant in any of the studies.

### 3.6. Risk of Bias

The risk of bias for each study was assessed using the Cochrane RoB 2.0 tool, considering the five domains and overall risk. Most of the studies were flawed in terms of the sample randomization method, and none of the studies offered a detailed description of allocation concealment. All studies had a high risk of bias with regards to domain 2, which addresses deviations from the intended intervention and includes “blinding of the participants and assessors”, as blinding was hindered by the use of Kinesio taping. A low risk of bias was found in all studies with regards to domain 3, which addresses missing outcome data. For domain 4 (“measurement of outcome”), however, a high risk of bias was found for all studies, as edema was measured by hand, which can lead to variations, and the use of different elastic bandages and types of application may have influenced the patient response. A low risk of bias was found for domain 5 (selective reporting), as the primary outcomes of clinical importance were reported.

## 4. Discussion

The qualitative results of this study suggest that KT can reduce postoperative morbidity following facial fracture surgeries. Due to the heterogeneity of the data, meta-analysis was not possible for all the three variables of interest. However, meta-analysis was performed for the edema variable with two viable articles. The results demonstrated a better performance in the groups submitted to the proposed intervention method (KT), as seen in the graphs, which enabled visualizing the results, favoring (without statistical significance), in terms of its effects, the KT groups over the control groups in the two postoperative assessment times considered (Figure 2 and Figure 3).

No previous review was found to investigate the use of Kinesio therapy following facial trauma surgery, which justifies the present review. Systematic reviews involving the analysis of clinical studies that employed KT following third molar surgeries concluded that the method is promising in terms of reducing edema [33,34,35]. 

In the postoperative period of facial fracture surgery, a certain degree of morbidity is expected, especially edema, pain and limited mouth opening, which can affect the social, financial and mental wellbeing of patients [26]. Diverse methods are employed with the aim of avoiding or reducing the occurrence of these signs and symptoms related to the inflammatory process and optimizing the recovery process. Thus, there is a continual search for accessible measures, such as Kinesio taping, which was the object of the present systematic review.

Preemptive analgesia is employed with the aim of preventing peripheral and central sensitization, attenuating (or ideally preventing) the increase in postoperative pain sensation. Among the pharmacological methods used for this purpose, the literature describes local anesthetics and/or opioids, as well as steroid and non-steroidal anti-inflammatory drugs (NSAIDs) [36]. These latter two medications are also commonly used to control postoperative edema and pain [6,7,8]. The combined administration of corticosteroids with NSAIDs has been considered to reduce postoperative inflammation. Cortisol and synthetic cortisol analogs can interfere with inflammatory processes, thus suppressing characteristic symptoms, while NSAIDs act by regulating the synthesis of prostaglandins, closely associated with pain and inflammation [37]. KT has proven to be a viable option that can assist in diminishing the duration of use and possible side effects of these medications. Among the studies included in the present review, Krishnamurthy et al. (2021) administered 1000 mg hydrocortisone half an hour prior to anesthetic induction and used NSAIDs in the postoperative period that can exert an influence on the inflammatory process, such as 1000 mg paracetamol [23,24,30], 600 mg ibuprofen [23,24] and 50 mg diclofenac [31,32]. As seen in Table 2, the studies exhibit a diversity in medication patterns administered during the study period, spanning from preoperative to postoperative phases. 

Before applying KT, it is essential that the patient’s skin is clean, free from dirt, oil, or sweat. Additionally, in the presence of body hair, such as facial hair, it is crucial to remove it to ensure optimal adherence of the therapeutic bandage [38]. During application, Kinesio tape is gently elongated, stretching the skin. The strips return to their original length after use. This mechanism causes tension in the skin and results in folds or villosities responsible for increasing the interstitial space between the skin and connective tissue, promoting greater fluidity of lymph and blood. Inflammatory fluids are directed from areas of greater pressure to areas of lower pressure under the influence of the elastic bandage, consequently contributing to a reduction in edema [39,40,41].

All studies (RCTs) in the present systematic review reported results with trends and positive effects on edema control in the groups submitted to KT. Maximum edema generally occurs on the second or third day of the postoperative period, as seen in the control groups. However, peak edema was reached in the first day of the postoperative period in the experimental groups involving Kinesio taping. When analyzing the data related to edema measurements in Table 3, an early reduction is observed in comparison to the groups that did not undergo the therapy [26,27,29,30,31]. The 60% reduction in edema found in the first two days of the postoperative period in the intervention groups was attributed to the properties of stretching, thickness, adhesivity and elasticity of the tape during application [23,24,26,31].

Mouth opening can be compromised in the postoperative period due to tissue trauma. Limited mouth opening characterizes trismus. A significantly better performance in the maintenance of mouth opening in the KT group over the control group was only reported in the study conducted by Bhushan et al. (2022), whereas an improvement was found in the other studies that addressed trismus, but the assessment was subjective [23,24,26,31]. For this assessment to be more precise, differences between the baseline and postoperative measurements should be considered [35]. This improvement may be due to the faster resolution of edema, which lessens skin tension. However, the placebo effect cannot be discarded [24,26].

KT did not exert an influence on the pain outcome, as no significant difference between groups was found for this variable, which is in agreement with data described in previous studies [42,43]. Moderate or mild pain (VAS < 5) was found at all predetermined assessment times in both groups, except in the study conducted by Deleme et al. (2021) [30]. This finding can be attributed to the administration of analgesics used in the patients participating in the included studies.

Firoozi et al. (2022) suggested that the effect of KT on pain control may be related to the “gate control” theory [44], which states that afferent fibers of sensory neurons of touch have a greater diameter and conduction velocity compared to afferent fibers corresponding to the sensory neurons of pain. Thus, by stimulating afferent receptors of touch with KT, the transmission of pain can be inhibited or diminished. Moreover, low-threshold skin mechanoreceptors that reduce the sensation of pain are activated due to the elastic nature of the tape, which promotes the stretching and elevation of the skin during movement [23]. 

The absence of homogeneity in the data impeded quantitative analysis of trismus and pain. Not all studies included in this review used similar ways to express the data (mean and mean with standard deviation) [23,26,27,29,30,31], described the data clearly [24,26] or even performed an investigation of these variables [30,32].

KT is considered uncomplicated and assists in the healing process following maxillofacial surgeries [35]. Observing the methodology of the clinical trials included in this review, however, there is an evident need for the standardization of methods from the application of the tape to measures used in the determination of changes in edema. The studies included used different methods for measuring edema. The two main methods were five and three linear measurements, but different anatomic reference points were considered, and, consequently, the studies did not achieve numerically similar values. There is no restriction with regards to the use of KT by healthcare providers in the field of oral-maxillofacial care, but training enables knowledge and the proper application of the method.

The disadvantages of KT include the potential for allergic reactions due to the adhesive that binds the fabric to the skin. Additionally, there may be challenges in social interaction due to the presence of the tape on the face. Skin irritations may also arise from perspiration. In cases where undesirable side effects occur, the application should be discontinued, and the treatment halted. The contraindications for KT include open wounds, skin infections, allergies to the adhesive used in the tape and compromised circulation at the surgical procedure site [35,38]. However, the studies included in this review did not describe the occurrence of associated adverse effects in the study participants. Moreover, they found a positive impact of KT on tissue response and a reduction in edema following facial trauma surgery.

The short follow-up period (seven days) in most of the studies included [23,24,29,30,31] facilitates the accompaniment of the patients, reduces the dropout rate and, consequently, improves the quality of the pooled results. In the studies considered, however, blinding was only possible for patients that were not aware that KT was employed as therapy. In contrast, blinding was not possible for the examiners due to the nature of the application of the elastic bandages, characterizing a single-blind study, which could bias the results.

As limitations of this study, the heterogeneity of the data, as mentioned earlier, poses challenges for conducting a meta-analysis for all variables of interest. Additionally, the lack of standardized methods among studies, especially in edema measurement, and variation in anatomical reference points contribute to the difficulty in drawing definitive conclusions. Furthermore, the absence of a unified protocol for KT application and different follow-up durations among studies adds complexity to result interpretation.

Despite these limitations, the systematic review provides valuable insights into the potential benefits of Kinesio tape as a complementary therapy in postoperative facial surgery. Qualitative analysis reveals positive outcomes in edema control. The study contributes to the existing literature by addressing a gap in research related to the use of KT in the context of facial trauma surgery.

The considered studies were classified as having a high risk of bias due to the lack of sample randomization and blinding, as well as the manual method used for edema measurement. These factors led to a reduction in the methodological quality score when using the Cochrane RoB 2.0 assessment method [45]. Therefore, future research in this domain should prioritize methodological standardization to enhance result reliability and reduce bias. Improvements in the randomization process, clearer descriptions of applied methods, and efforts toward double-blinding strategies (both for patients and assessors) are recommended. Objective measures for facial edema, such as advanced imaging acquisition techniques, could be explored to provide more precise and quantifiable data. Additionally, systematic approaches to medication protocols, both pre- and postoperatively, should be considered to eliminate potential measurement errors associated with anti-edema medications.

## 5. Conclusions

In summary, the qualitative findings of this systematic review suggest that Kinesio tape can effectively reduce postoperative morbidity following facial fracture surgeries, particularly in edema control. The overall analysis supports the straightforward nature of KT, making it relevant as a complementary therapy due to the benefits it offers to patients.

## Figures and Tables

**Figure 1 jcm-13-00997-f001:**
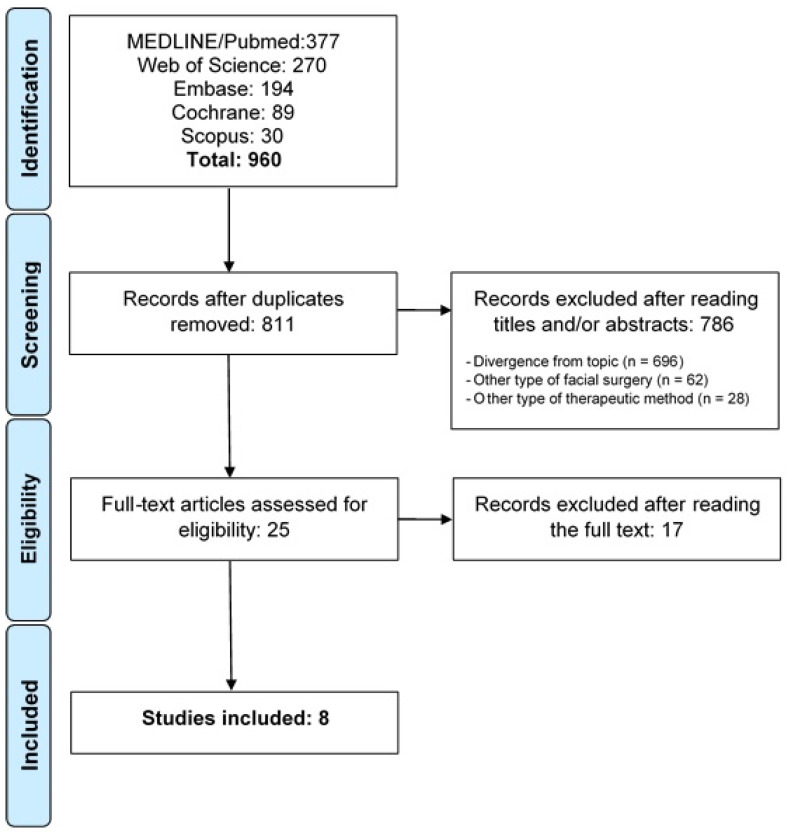
Flow chart of study selection process in accordance with PRISMA guidelines.

**Figure 2 jcm-13-00997-f002:**
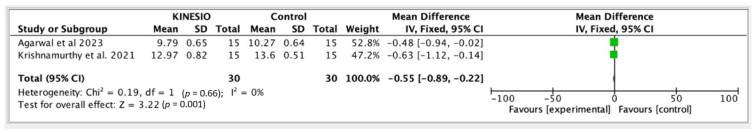
Meta-analysis for the second postoperative day [26,27].

**Figure 3 jcm-13-00997-f003:**
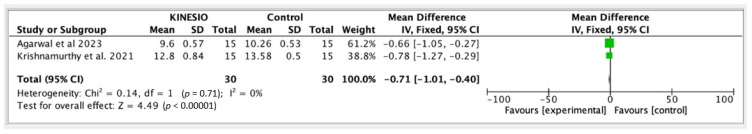
Meta-analysis for the third postoperative day [26,27].

**Table 1 jcm-13-00997-t001:** Methodological and participant characteristics of the randomized clinical trials included in the systematic review.

Investigator	Country	Fracture Region (Inclusion Criteria)	Post-Op Follow-Up (Days)	Age		Sample Size		Gender			
				KT Group	Control Group	KT Group	Control Group	KT Group		Control Group	
				Mean ± SD (Years)	Mean ± SD (Years)	Participants (n)	Participants (n)	Male (n)	Female (n)	Male (n)	Female (n)
Ristow, Hohlweg-Majert et al., 2013 [23]	Germany	Mandible	7	43.8 ± 20.7	42.5 ± 16.7	13	13	5	8	6	7
Ristow, Pautke et al., 2013 [24]	Germany	Zygomatico-Maxillary Complex	7	NR	NR	15	15	NR	NR	NR	NR
Ristow et al., 2014 [31]	Germany	Zygomatico-Maxillary Complex and Mandible	7	NR	NR	56	56	NR	NR	NR	NR
Tyndorf et al., 2016 [32]	Poland	Orbital margin and Mandible	10	NR	NR	22	19	16	6	12	7
Deleme, Aljubory 2021 [30]	Iraq	Mandible	7	NR	NR	10	10	8	2	9	1
Krishnamurthy et al., 2021 [26]	India	Zygomatico-Maxillary Complex	7	33	35	15	15	15	0	15	0
Bhushan, Sharma 2022 [29]	India	Zygomatico-Maxillary Complex	7	30.20 ± 8.76	31.87 ± 10.50	15	15	13	2	14	1
Agarwal et al., 2023 [27]	India	Mandible	5	NR	NR	15	15	NR	NR	NR	NR

Abbreviations: KT, Kinesio taping; SD, standard deviation; NR, not reported.

**Table 2 jcm-13-00997-t002:** Characteristics of the Kinesio taping used, medication protocol adopted and swelling measurements.

Investigator	KT				Pre-Medication in Both Groups	Post-Medication in Both Groups
	Type of KT	Technique Applied	Tape Size	Time with the Tapes (Days)		
Ristow, Hohlweg-Majert et al., 2013 [23]	K-Active Kinesiology Tape Classic	A	0.5 cm each strip	5	NR	1000 mg paracetamol IV, 2 times per day for 3 days and 600 mg ibuprofen orally (1st day: 600 mg ibuprofen 3 times per day; 2nd day: 600 mg ibuprofen 2 times per day; 3rd day: 600 mg ibuprofen 1 time per day; 4th day: 600 mg ibuprofen 1 time per day). Antibiotic treatment was continued with ampicillin and sulbactam IV, 3 times per day for 3 days
Ristow, Pautke et al., 2013 [24]	K-Active Kinesiology Tape Classic	B	1.5 cm each strip	5	Single shot of 2000 mg/1000 mg ampicillin/sulbactam	1000 mg paracetamol IV, 2 times per day for 3 days; and 600 mg ibuprofen (1st day: 600 mg ibuprofen 3 times per day, 2nd day: 600 mg ibuprofen 2 times per day, 3rd day: 600 mg ibuprofen 1 time per day, 4th day: 600 mg ibuprofen 1 time per day)
Ristow et al., 2014 [31]	K-Active Kinesiology Tape Classic	A	1.5 cm each strip	5	Single shot of ampicillin/sulbactam, 2000 mg/1000 mg	50 mg diclofenac every 8 h for 3 days
Tyndorf et al., 2016 [32]	Nitto Crown brand	C	NR	NR	NR	NR
Deleme, Aljubory 2021 [30]	Kinesio Tex	D	0.5 cm each strip	5	NR	1 g ceftriaxone every 24 h IV daily for three days for both groups, and 75 mg diclofenac sodium IM every 24 h daily for two days then giving diclofenac suppositories for subsequent days when needed for non KT group only. KT group given 1 g paracetamol IM
Krishnamurthy et al., 2021 [26]	NR	E	1.5 cm each strip	NR	100 mg hydrocortisone half an hour prior induction	No corticosteroid or proteolytic enzyme was administered
Bhushan, Sharma 2022 [29]	K-Active Kinesiology Tape Classic	B	NR	5	NR	NR
Agarwal et al., 2023 [27]	Dynaplast, Johnson & Johnson	F	Three longitudinal slits with four strips 1 cm wide	5	NR	750 mg paracetamol, 500 mg amoxicillin and 0.12% chlorhexidine mouth washes were given for 5 days *

Abbreviations: KT, Kinesio taping; NR, not reported; IV, intravenously; IM, intragluteal injection. A: The tape was cut into 3 equal strips, placed lightly on the lymph node area to which the drainage was being directed (supraclavicular nodes). The patient was moved into a stretched position. Tails were placed on the skin with slight tension (20%). Placement of the lymphatic stripes was directed at the appropriate lymphatic duct crossing the cervical, submental, mandibular, submandibular, preauricular, and parotid nodes to the area of maximum swelling. B: The tape was cut into 3 equal strips, placed lightly on the lymph node area to which the drainage was being directed (supraclavicular nodes). The base was placed slightly above the lymph node area to which the drainage is being directed (supraclavicular nodes). The patient was moved into a stretch position. Tails were placed on to the skin with slight tension (20%). Placement of the lymphatic stripes was directed at the appropriate lymphatic duct crossing the cervical, submental, mandibular, submandibular, preauricular and parotid nodes, crossing the zygomatic arch, reaching the infraorbital rim and frontozygomatic suture surrounding the lower eye. C: The tape was cut into 5 equal strips, placed lightly in the region of the lymph nodes where the drainage was being directed (supraclavicular nodes); in this region, the base from which the 5 strips originate was kept intact. Placement of the lymphatic strips was directed to the appropriate lymphatic duct crossing the cervical, submental, mandibular, submandibular, preauricular, and parotid nodes. D: Tape dimension was measured for each case from the ear lobule to the point of mandibular mid line. The tape was sectioned into 2 equal strips and their tail was rounded down. The patient was moved into a stretched position. Tails were placed on the skin with slight tension (20%). E: The length was individually measured for each patient from the clavicle to the highest point of the swelling. F: The adhesive tape was then applied from the base of the mandible in the submandibular region (a fixed point) and covered the area below the ear lobe towards the entire labial commissure. * The posology was not informed.

**Table 3 jcm-13-00997-t003:** Measurement method, follow-up time and swelling measurements.

Swelling										
Investigator	Measurement Methods	Form of Presentation	KT Group							
			T-1	T0	1st Day	2nd Day	3rd Day	5th Day	7th Day	10th Day
Ristow, Hohlweg-Majert et al., 2013 [23]	5 lines, type A	Mean sum ± SD (mm)	62.8 ± 4.0	66.0 ± 4.5	NR	NR	NR	NA	NR	NA
Ristow, Pautke et al., 2013 [24]	5 lines, type A	Mean sum ± SD (mm)	62.1 ± 3.8	66.0 ± 3.9	NR	NR	NR	NA	NR	NA
Ristow et al., 2014 [31]	5 lines, type A	Mean sum ± SD (mm)	62.0 ± 4.4	65.2 ± 4.5	65.2 ± 4.6	63.5 ± 4.3	62.5 ± 4.2	NA	61.6 ± 4.2	NA
Tyndorf et al., 2016 [32] *	5 lines, type B	NR	NA	NA	NR	NR	NA	NR	NA	NR
Deleme, Aljubory 2021 [30]	3 lines, type A	Mean (mm)	NA	NA	9.8	NA	10.2	NA	9.45	NA
Krishnamurthy et al., 2021 [26] ^Ɨ^	5 lines, type A	Mean ± SD (mm)	12.79 ± 0.74	13.04 ± 0.62	13.10 ± 0.69	12.97 ± 0.82	12.80 ± 0.84	NA	12.73 ± 0.83	NA
Bhushan, Sharma 2022 [29]	5 lines, type A	Mean sum (mm)	64.20	68.16	67.20	66.06	65.16	NA	63.60	NA
Agarwal et al., 2023 [27] ^§^	3 lines, type B	Mean ± SD (mm)	10.54 ± 0.92	NA	NA	9.79 ± 0.65	9.60 ± 0.57	9.53 ± 0.55	NA	NA
**Investigator**	**Measurement Methods**	**Form of Presentation**	**Control Group**							
			**T-1**	**T0**	**1st Day**	**2nd Day**	**3rd Day**	**5th Day**	**7th Day**	**10th Day**
Ristow, Hohlweg-Majert et al., 2013 [23]	5 lines, type A	Mean sum ± SD (mm)	65.1 ± 2.8	67.5 ± 2.3	NR	NR	NR	NA	NR	NA
Ristow, Pautke et al., 2013 [24]	5 lines, type A	Mean sum ± SD (mm)	63.5 ± 4.3	65.8 ± 3.7	NR	NR	NR	NA	NR	NA
Ristow et al., 2014 [31]	5 lines, type A	Mean sum ± SD (mm)	63.0 ± 4.4	65.3 ± 4.3	67.3 ± 4.7	67.6 ± 5.0	67.0 ± 5.0	NA	64.8 ± 4.8	NA
Tyndorf et al., 2016 [32] *	5 lines, type B	NR	NA	NA	NR	NR	NA	NR	NA	NR
Deleme, Aljubory 2021 [30]	3 lines, type A	Mean (mm)	NA	NA	11.2	NA	10.8	NA	11.55	NA
Krishnamurthy et al., 2021 [26] ^Ɨ^	5 lines, type A	Mean ± SD (mm)	13.19 ± 0.56	13.33 ± 0.50	13.56 ± 0.57	13.60 ± 0.51	13.48 ± 0.50	NA	13.25 ± 0.54	NA
Bhushan, Sharma 2022 [29]	5 lines, type A	Mean sum (mm)	64.76	68.16	69.10	69.06	68.40	NA	65.83	NA
Agarwal et al., 2023 [27] ^§^	3 lines, type B	Mean ± SD (mm)	10.76 ± 1.19	NA	NA	10.27 ± 0.64	10.26 ± 0.53	10.22 ± 0.72	NA	NA

Abbreviations: KT, Kinesio taping; SD, standard deviation; T-1, pre-operative; T0, immediate post-operative; NR, not reported; NA, not available. Five lines, type A: line A, most posterior point of the tragus to the most lateral point of the lip commissure; line B, most posterior point of the tragus to the pogonion; line C, most posterior point of the tragus to the lateral canthus of the eye; line D, lateral canthus of the eye to most inferior point of angle of the mandible; line E, most inferior point of the angle of the mandible to the middle of the nasal bone. Five lines, type B: line A, point alare to ectoconchion; line B, point alare to zygien; line C, point orbitale to gonion; line D, point gnathion to otobasion inferior; line E, point alare to gnathion. Three lines, type A: line A, tragus to midline of the pogonion; line B, tragus to lip commisure; line C, was from lateral canthus of the eye to the angle of the mandible. Three lines, type B: line A, tragus to lip commisure; line B, was from lateral canthus of the eye to the angle of the mandible; line C, from the angle of the mandible to the lip commisure. * The data are exposed in graphs; it was not possible to extract exact values. ^Ɨ^ Mean of the lines used in the measurement ± SD (5 lines). ^§^ Mean of the lines used in the measurement ± SD (3 lines).

## Data Availability

Not applicable.

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
