# Peer review of "Effectiveness of Elastic Therapeutic Tape in Reducing Edema, Pain and Trismus following Surgery for Facial Fractures: A Systematic Review and Meta-Analysis"

_jcm, 2024, doi:10.3390/jcm13040997_

Round 1
Reviewer 1 Report
Comments and Suggestions for Authors
Dear authors,
Interesting subject but there is a discrepancy between the very detailed presentation of results and the scarcity of discussions. Please develop the section Discussions more based on the results in order to better understand the signification of these results in the context of knowledge.
In the methodology, please specify the exclusion criteria used that led to removal of 97% of articles! (786 from 811)
Could means the title Dor from the section 3.5, Pain ? Maybe is in portuguese?
Author Response
Dear Reviewer,
Thank you very much for taking the time to review this manuscript and for your thoughtful considerations aimed at enhancing the credibility and reliability of our work. Below is a detailed response to your comments, with corresponding revisions/highlights in red made in the resubmitted manuscript file.
Comments 1: Interesting subject but there is a discrepancy between the very detailed presentation of results and the scarcity of discussions. Please develop the section Discussions more based on the results in order to better understand the signification of these results in the context of knowledge.
Response 1: Modifications were made in the discussion, incorporating a more in-depth analysis of the obtained results. This involved emphasizing their significant implications and establishing a more direct connection to the study's objectives and relevance.
Comments 2: In the methodology, please specify the exclusion criteria used that led to removal of 97% of articles! (786 from 811)
Response 2: The criteria that led to the exclusion of the 786 articles have been added and specified in Figure 1. Flow chart of study selection process in accordance with PRISMA guidelines (page 4).
- Topic divergence (n = 696)
- Other types of facial surgery (n = 62)
- Other types of therapeutic methods (n = 28)
Comments 3: Could means the title Dor from the section 3.5, Pain ? Maybe is in portuguese?
Response 3: The term "Dor" refers to "Pain". During the translation into English, the term was kept in Portuguese, but the correction was made in the file (page 10, topic 3.5).
Reviewer 2 Report
Comments and Suggestions for Authors
Valued authors,
thank you for writing this systematic review and meta-analysis analysing the effectiveness of kinesio taping in facial trauma. As you have noticed, many papers have already been published on this topic. The main problem is always to achieve homogeneity of the data examined. This is reflected in the number of excluded articles (=952). In principle, it is difficult to make a significant statement about the volume of swelling because it depends on many factors, such as the type of fracture pattern, duration of the operation, dependence on the surgeon skills, the constitutional type of the patient, the application of different elastic bandages and types, and so on. Therefore, it is not surprising that in this article a value of p<0.10 can already be viewed as statistically significant. That's why I wouldn't write about statistically significant results, but rather about tendencies and effects.
The article is well structured. For my part, I would not mention any search terms in section 2.3. Also leave out the listed journals on lines 93-95, as they do not provide much more information. It is also questionable, whether table 4 must be displayed. Also because, out of 8 studies, the results in 4 were not recorded or were not available.
The title on line 263 should be Dolor and not Dor.
I would also leave out figure 4. Graphically, it is even more emphasized that the included articles are very biased. Instead of this figure, I would describe this fact in the discussion.
In summary, I can say that this paper highlights many of the problems with such a topic.
Author Response
Dear Reviewer,
Thank you very much for taking the time to review this manuscript and for your thoughtful considerations aimed at enhancing the credibility and reliability of our work. Below is a detailed response to your comments, with corresponding revisions/highlights in red made in the resubmitted manuscript file.
Comments 1: In principle, it is difficult to make a significant statement about the volume of swelling because it depends on many factors, such as the type of fracture pattern, duration of the operation, dependence on the surgeon skills, the constitutional type of the patient, the application of different elastic bandages and types, and so on. Therefore, it is not surprising that in this article a value of p<0.10 can already be viewed as statistically significant. That's why I wouldn't write about statistically significant results, but rather about tendencies and effects.
Response 1: The term "statistically significant" was changed in some sections of the discussion to address the reviewer's suggestion. This adjustment was particularly important due to the acknowledgment that a statistical difference does not necessarily imply a clear clinical impact (page 10, paragraph 1 of the discussion, line 273/ page 11, paragraph 2, line 312-313). Additionally, modifications were made in the discussion, involving a more thorough analysis of the obtained results. This included emphasizing their significant implications and establishing a more direct connection to the study's objectives and relevance.
Comments 2: For my part, I would not mention any search terms in section 2.3.
Response 2: The search terms (search strategy) have been removed, following the reviewer's guidance.
Comments 3: Also leave out the listed journals on lines 93-95, as they do not provide much more information.
Response 3: The listed journals have been removed, following the reviewer's guidance.
Comments 4: It is also questionable, whether table 4 must be displayed. Also because, out of 8 studies, the results in 4 were not recorded or were not available.
Response 4: The authors agree with the reviewer's suggestion; consequently, Table 4 has been excluded due to the absence of available data. The information provided in the corresponding paragraphs, "3.4. Trismus" and "3.5. Pain", is deemed sufficient for comprehension.
Comments 5: The title on line 263 should be Dolor and not Dor.
Response 5: The term "Dor" refers to "Pain". During the translation into English, the term was kept in Portuguese, but the correction was made in the file (page 10, topic 3.5).
Comments 6: I would also leave out figure 4. Graphically, it is even more emphasized that the included articles are very biased. Instead of this figure, I would describe this fact in the discussion.
Response 6: Figure 4 has been removed in accordance with the reviewer's suggestion. It is important to note that in the results section, the topic "3.6. Risk of bias" has been appropriately modified, and this aspect has been incorporated into the discussion (page 12, paragraph 6, line 386-389).
Reviewer 3 Report
Comments and Suggestions for Authors Dear Authors,It was a pleasure to read your article. I believe your paper might be
interesting to readers from the clinical field. Your paper is well
written and organized.
However, there are some scopes to improve the quality of the
manuscript. The reviewer would like to suggest the following
revision in the manuscript.
The aim of this technical note "Effectiveness of elastic
therapeutic tape on the reduction in edema, pain and trismus
following surgery for facial fractures: a systematic review and
meta-analysis" was to evaluate the effectiveness of elastic tape
Kinesio Taping (KT) in reducing postoperative morbidity in facial
fractures surgeries.
Minor editing of English language required
Punctuation should be corrected.
Standardize text structure and alignment according to guidelines.
Abstract:
p value should be written in italics.
Introduction
line 51 - add new reference: Pławecki, P.; et al. Kinesio Taping as
an Adjunct Therapy in Postoperative Care after Extraction of Impacted
Third Lower Molars—A Randomized Pilot Study. J. Clin. Med. 2023, 12,
2694. https://doi.org/10.3390/jcm12072694
Materials and Methods
Add inclusion and exclusion criteria as separate subsections.
p value should be written in italics.
Results
lines 138-139 - skip to next page
lines 177-178 skip to next page
Prepare tables according to the guidelines.
table 2 - skip to page 6
The quality of the figure 3 is poor. Increase quality. Figure 3
should be larger.
Discussion
Describe in what situations kinesio tape cannot be applied?
What are the contraindications?
Add to the discussion that pre-treatment prevention
(antibiotics and non-steroidal anti-inflammatory drugs)
may result in better healing, less pain and swelling.
Please describe more clearly the limitations of this study, not
just the studies you used. Add the advantages of the study.
Discuss future opportunities for researchers and research directions.
Conclusions
No conclusions. Please add.
Add a table with abbreviations before references.
Reconsider after major revision Comments on the Quality of English Language
Minor editing of English language required
Author Response
Dear Reviewer,
Thank you very much for taking the time to review this manuscript and for your thoughtful considerations aimed at enhancing the credibility and reliability of our work. Below is a detailed response to your comments, with corresponding revisions/highlights in red made in the resubmitted manuscript file.
Comments 1: Minor editing of English language required.
Response 1: The term "Dor" refers to "Pain". During the translation into English, the term was kept in Portuguese, but the correction was made in the file (page 10, topic 3.5).
Comments 2: Standardize text structure and alignment according to guidelines.
Response 2: I assert that the structure has been organized, and the alignment has been done in accordance with the guidelines.
Comments 3:
Abstract:
p value should be written in italics.
Response 3: The p-value should be written in italics (page 1, lines 26-27).
Comments 4:
Introduction:
line 51 - add new reference: PÅ‚awecki, P.; et al. Kinesio Taping as an Adjunct Therapy in Postoperative Care after Extraction of Impacted Third Lower Molars—A Randomized Pilot Study. J. Clin. Med. 2023, 12, 2694. https://doi.org/10.3390/jcm12072694
Response 4: We agree with the reviewer's suggestion and have added the recommended study to our list of references to enhance the robustness of our manuscript (page 2, line 51).
Comments 5:
Materials and Methods
Add inclusion and exclusion criteria as separate subsections.
Response 5: The suggested changes have been implemented (page 2, lines 64 and 72).
Comments 6:
Materials and Methods
p value should be written in italics.
Response 6: The p-value should be written in italics (page 2, line 97).
Comments 7:
Results
Lines 138-139 - skip to next page/lines 177-178 skip to next page/Prepare tables according to the guidelines/table 2 - skip to page 6
Response 7: Due to changes in the text, the lines are no longer corresponding, however, the manuscript has been carefully formatted to address these highlighted points. Additionally, the tables adhere to the guidelines established for authors.
Comments 8:
Results
The quality of the figure 3 is poor. Increase quality. Figure 3 should be larger.
Response 8: The quality of the figure has been enhanced.
Comments 9:
Discussion
Describe in what situations kinesio tape cannot be applied? What are the contraindications?
Response 9: The contraindications have been added (page 12, paragraph 2, lines 363-365).
Comments 10:
Discussion
Add to the discussion that pre-treatment prevention (antibiotics and non-steroidal anti-inflammatory drugs) may result in better healing, less pain and swelling.
Response 10: We have accepted the reviewer's suggestion, and additional information has been incorporated (page 10, paragraph 4 of the discussion, lines 286-295).
Comments 11:
Discussion
Please describe more clearly the limitations of this study, not just the studies you used. Add the advantages of the study.
Response 11: The limitations and advantages have been incorporated into the body of the text (page 12, paragraph 4 and 5, lines 375-385).
Comments 12:
Discussion
Discuss future opportunities for researchers and research directions.
Response 12: The paragraph containing suggestions for future research has been altered and rewritten (page 12, paragraph 6, lines 386-397).
Comments 13:
Conclusions
No conclusions. Please add.
Response 13: The "Conclusions" section has been added, along with its respective paragraph (page 13, paragraph 1, lines 403-406).
Comments 14:
Conclusions
Add a table with abbreviations before references.
Response 14: A table containing abbreviations has been inserted before the references, as suggested by the reviewer (page 13, line 423).